# Master-Key Regulators of Sex Determination in Fish and Other Vertebrates—A Review

**DOI:** 10.3390/ijms24032468

**Published:** 2023-01-27

**Authors:** Arie Yehuda Curzon, Andrey Shirak, Micha Ron, Eyal Seroussi

**Affiliations:** 1Institute of Animal Science, Agricultural Research Organization, Rishon LeTsiyon 75288, Israel; 2Robert H. Smith Faculty of Agriculture, Food and Environment, Hebrew University of Jerusalem, Rehovot 76100, Israel

**Keywords:** sex-determination, master key regulator, sex-ratio, mono-factorial, hybridization, validation

## Abstract

In vertebrates, mainly single genes with an allele ratio of 1:1 trigger sex-determination (SD), leading to initial equal sex-ratios. Such genes are designated master-key regulators (MKRs) and are frequently associated with DNA structural variations, such as copy-number variation and null-alleles. Most MKR knowledge comes from fish, especially cichlids, which serve as a genetic model for SD. We list 14 MKRs, of which *dmrt1* has been identified in taxonomically distant species such as birds and fish. The identification of MKRs with known involvement in SD, such as *amh* and *fshr*, indicates that a common network drives SD. We illustrate a network that affects estrogen/androgen equilibrium, suggesting that structural variation may exert over-expression of the gene and thus form an MKR. However, the reason why certain factors constitute MKRs, whereas others do not is unclear. The limited number of conserved MKRs suggests that their heterologous sequences could be used as targets in future searches for MKRs of additional species. Sex-specific mortality, sex reversal, the role of temperature in SD, and multigenic SD are examined, claiming that these phenomena are often consequences of artificial hybridization. We discuss the essentiality of taxonomic authentication of species to validate purebred origin before MKR searches.

## 1. Sex Determination

Sex determination (SD) is a fundamental biological process that drives the optimal sex ratio for reproduction and for defective allele purging in natural populations [1,2,3]. In birds and mammals, SD mechanisms are mainly genetic (GSD), whereas lower vertebrates show a variety of SD mechanisms, such as genetic, environmental (ESD), social factors, and their combinations [4].

Mainly performed on fish species, most studies show that a single genetic sequence, which could be a gene or any other genomic element, triggers SD for each species (Table 1), resulting in a dichotomic segregation of sexes and an initial even sex ratio. Such factors are designated master-key regulators (MKRs) of SD [5]. However, some researchers prefer to denote only a protein-producing gene as the causative regulator and do not include other genomic elements, which could be considered a terminology problem, e.g., *amhΔy* vs. *amhy* in Nile tilapia (discussed in Section 4) [6]. Many studies have mapped a region for SD in different species, and suggest multiple candidates from the mapped region. However, this review considers only those candidate MKRs that were suggested after isolating a specific DNA sequence variation, which was then confirmed by recombination boundaries, multiple species conservation, or functional studies (Table 1). MKRs are not universal, and closely related species may utilize different MKR genes for SD [7,8,9]. In some species, sex ratios in adults are significantly influenced by temperature during early development. However, as shown in avian species, ratios can also be modulated by sex-specific survival rates [10]. Thus, deviation from the equal sex ratio is not always an indication of the number and type of factors involved in SD for an examined species, since sex-specific mortality can mimic environmental SD [11,12].

Two common scenarios have been observed for vertebrate MKR genes. In a minority of cases, the MKR is a sex-specific gene with two different allele variants for males and females. However, in the majority of cases, MKRs are associated with copy-number variation (CNV), in which a unique variant evolved (on the Y or W chromosome) into an SD regulator [9]. When the MKR is present in a single copy, it may be regarded as a CNV involving a null allele, as in the case of *dmrt1* in birds (Table 1). The existence of common MKRs in taxonomically distant species indicates that the number of MKRs is limited to a set of factors which belong to a conserved regulatory SD pathway [13,14]. This pathway controls the SD cascade, and triggers primary differentiation of the bi-potential gonad up to the stage of the sex-specific pattern of steroid hormone synthesis [15,16]. An extremely well-observed example of this is in the paralogs of *amh* and *dmrt1*, which serve as MKRs of SD in many distant fish species (Table 1): an *amh* copy was found to be the initiator of only the XY/XX SD mechanism, whereas *dmrt1* was found to initiate both WZ/ZZ and XY/XX SD mechanisms. Moreover, currently, only two additional MKR genes have been found as the initiators of WZ/WW mechanisms: *hsd17b* and *banf2* (Table 1). The fact that specific MKRs initiate a certain type of SD mechanism i.e., either the XX/XY or the ZZ/WZ, or both systems, is an indication of the existence of conserved specialization and hierarchy of factors in the SD pathway.

The model of interaction of GSD and ESD in lower vertebrates is based on two principles. First, the gonads appear to be morphologically identical in both sexes (bi-potential gonad), but through sex-hormone synthesis, the MKR induces differentiation into the ovary or testis [17]. This process of puberty can take place at different ages, depending on the species. Second, depending on the type of hormones secreted, which could be affected by environment, morphological differentiation might be terminated or altered completely, causing sex-reversal [17,18]. Furthermore, the genes of SD MKRs are frequently expressed beyond the period of embryonic development, indicating that they are not involved only in SD initiation, but also in sex development and maintenance. For example, the human MKR, *SRY,* is expressed in different human tissues up to adolescence, independent of gonadal hormone levels [19]. There are also surprising discoveries of MKRs, such as the immune-related gene (*sdY*) of salmonids [20,21], showing that a complex regulation mechanism of SD may have evolved [22]. Yet, a recent study has found *sdY* integration with the classical SD cascade by interaction with *foxl2*, which is associated with ovarian development [23]. However, many central factors of the SD pathway, such as *sox9*, *wt1*, *dax1*, *sf1*, *wnt4*, *foxl2* and *cyp19a1a*/*b* were not observed to have a natural MKR function in any fish species studied so far (Table 1). This is puzzling because transgenic experiments have demonstrated that knockouts in some of these central genes may lead to sex reversal [24,25]. It is possible that these genes are conserved because of their critical role in development, and thus sex specific variation might adversely affect their function.

Fish demonstrate a wide variety of SD MKRs and are, therefore, an excellent model for the study of SD and the sex differentiation cascade. Moreover, SD of closely related species can be controlled by different MKRs [26]. In such cases, interactions between different MKRs may be examined in the following generations, if the interspecific hybrids are viable [27]. Many fish have large populations with short generation intervals, thus facilitating effective genetic studies. Particularly, tilapia has been used in many SD studies to investigate the diversity of SD genes and their functional characteristics [28,29,30].

Environmental and social factors have been shown to be involved in SD of many low vertebrate species, and can be dominant over genetic factors [31]. However, in some cases, cryptic genetic factors may be involved together with environmental SD. For example, in the protandrous hermaphrodite gilthead seabream (*Sparus aurata*), a strong quantitative trait locus (QTL) was detected for SD, which does not overlap with a QTL for weight [32]. It may be speculated that in such species, progeny that are initially all-males or all-females, segregate for a cryptic genetic SD system that contributes to the propensity for sex reversal [33]. In addition, high temperatures have been shown to affect SD, resulting in skewed sex ratios [34]. However, sex-specific larval and adult mortality may also contribute to skewed sex ratios, which is not always taken into consideration in association studies [11,12]. Specifically in *O. niloticus* and *O. aureus*, high incubation temperatures (34–37 °C) may cause female to male sex-reversal [34]. It was shown that *O. aureus* is sensitive to temperature treatment, whereas *O. niloticus* stocks demonstrate a lesser degree of sensitivity to temperature sex-reversal induction [34]. Moreover, different levels of sensitivity were detected amongst families of the same population. A QTL for thermo-sensitivity was mapped to LG20 in the Stirling tilapia strain [35]. However, a later investigation of the appearance of multiple SD systems in different strains of *O. niloticus* raises the question of whether they stem from *O. niloticus* × *O. aureus* hybridization [36], and whether the temperature sensitivity originated from *O. aureus*. Our long-term study of reproductive activity and spawning of *O. aureus* in a natural temperature regime in Israel detected a significant decrease, or a complete retention of spawning in July to August, when temperatures exceed 29 °C [37]. In species with an XX/XY SD mechanism, this may be a natural mechanism to avoid the appearance of XX neo-males, and of Y chromosome loss, which could occur following subsequent reproduction. Recent analysis of purebred *O. niloticus* and of *O. aureus* populations of Uganda and Israel, respectively, with markers in MKR genes, detected 0–4% of sex reversed males. Therefore, the influence of temperature on SD in purebred tilapia populations, and perhaps in other species with GSD, is possibly restricted in natural conditions. Nonetheless, the detection of a QTL for temperature sensitivity on LG20 needs further investigation to determine the underlying genetic factors.

**Table 1 ijms-24-02468-t001:** Identification of Master Key Regulators (MKRs) of Sex Determination (SD) and their paralogs.

Human Ortho/Paralog	SD System	MKR	Organism/Species	Type ^1^	References/ Chr ^2^
*SRY-Box Transcription Factor 3* (*SOX3*)	XX/XY	*sox3Y* *sry*	*Oryzias dancena*, *O. marmoratus*, *O. profundicola*	SV	[26,38]/LG10
			mammals (most)	MSD	[39,40]/ChrY
*Anti-Müllerian Hormone Receptor 2* (*AMHR2*)	XX/XY	*amhr2y*	*Takifugu rubripes*, *Plecoglossus altivelis*	SV	[41,42]/Chr19
			*Perca flavescens*, *Phyllopteryx taeniolatus*, *Syngnathoides biaculeatus*, *Pangasiidae*	MSD	[43]/Chr4 [44]/Chr9 [45,46]/Chr4
no ortholog	XX/XY	*gsdfY*	*Oryzias luzonensis*, *Hippoglossus hippoglossus*, *Anoplopoma fimbria*	SV	[47]/LG12 [48]/Chr13 [49]/LG14
*Hydroxysteroid 17-Beta Dehydrogenase 1* (*HSD17B1*)	WZ/ZZ	*hsd17b1w*	*Seriola* genus, *Trachinotus anak*	SV	[50,51]/Chr24
*Bone Morphogenetic Protein Receptor Type 1B* (*BMPR1B*)	XX/XY	*bmpr1bbY*	*Clupea harengus*	MSD	[52]/Chr8
*Double-sex and Mab-3 Related Transcription Factor 1* (*DMRT1*)	XX/XY	*dmrt1Y* *dmrt1bY* *dmW dmrt1z*	*Oryzias latipes*	MSD	[53,54]/LG1 [55,56]/ChrW [57,58,59]
	XX/XY		*Scatophagus argus*		
	WZ/ZZ		*Xenopus laevis* (amphibians)	FSD	
	WZ/ZZ		*Gallus gallus* (birds), *Cynoglossus semilaevis*		
*Interferon Regulatory Factor 9* *(IRF9)*	XX/XY	*sdY*	Salmonids	MSD	[21]/BT18, RT01
*Growth Differentiation Factor 6* (*GDF6*)	XX/XY	*gdf6aY* *gdf6b*	*Nothobranchius furzeri*, *Astyanax mexicanus*	SV	[60,61]/ChrB
no ortholog	XX/XY	*zkY*	*Gadus morhua*	MSD	[62]/LG11
*Anti-Müllerian Hormone* (*AMH*)	XX/XY	*amhΔy*	*Odontesthes hatcheri*, *Oreochromis niloticus*, *Ophiodon elongates*, *Hypoatherina tsurugae*, *Esox lucius*, *Gasterosteus aculeatus*, *Sebastes schlegelii*, *Culaea inconstans*	MSD	[63] [64]/LG23 [65,66] [67]/LG24 [68,69] [70]/Chr10
*Barrier-* *to-Autointegration-* *Factor-like protein 2* (*BANF2*)	WZ/ZZ	*banf2w*	*Oreochromis aureus*, *Oreochromis urolepis hornorum, Pelmatolapia mariae*	FSD	[71]/LG3
*Follicle Stimulating Hormone Receptor* (*FSHR*)	XX/XY	*fshry*	*Mugil cephalus*	SV	[9]/LG9
*Folliculogenesis Specific BHLH Transcription Factor* (*FIGLA*)	XX/XY	*figla* *-like*	*Oreochromis mossambicus*, *Coptodon zillii*, *Sarotherodon galilaeus*	MSD	[36]/LG1

^1^ Sequence variation (SV) and structural sequence variation, including copy number variation (CNV) of male (MSD) or female (FSD) specific gene duplications. ^2^ Sex chromosome annotation for mapped MKRs (Chr).

## 2. Search for MKRs

The search for SD MKRs relies on linkage disequilibrium between genetic markers and the causative gene. Thus, genetic maps that are based on linkage between adjacent markers are being used for such studies [72,73]. Recently, genome sequence data are being used for genome-wide association studies (GWAS) [74,75]. Although sex is a categorical trait, which is usually controlled by a single gene, many studies analyze sex as a QTL [76,77,78,79]. Non-recombining regions are part of the characteristics of many sex chromosomes, which may cause extensive association with SD throughout the sex chromosome [80]. This has hampered fine-mapping studies of QTLs in tilapia [71]. Recent studies have shown that regional non-recombining blocks of purebred species can be broken down following hybridization, aiding in the fine genetic mapping of MKRs for SD [27,71,81,82]. Multiple sex determination loci that may segregate in a single strain may also contribute to the complexity of MKR searches [27,83]. Sex specific mortality [11,12] and sex reversal [34] also affect the sex phenotype, and may involve multiple genetic loci.

Although the traditional candidate gene approach may be used for MKR detection, it has been considered unproductive in comparison with GWAS [84]. Moreover, results of the candidate gene approach in the analysis of the genetic control of human diseases have been “woefully inadequate” [85]. Nevertheless, since the number of MKRs of SD seems to be limited, the identification of MKRs (Table 1) may indicate candidate genes for future searches for MKRs in additional species, using heterologous sequences without prior genetic mapping.

## 3. Identification of SD MKRs

The discovery of the mammalian *Sry* [39,40] uncovered an extra-numeral copy in males that evolved from *sox3* [86,87,88,89]. This was followed by the identification of other SD MKRs in *Oryzias latipes dmrt1* [53,54] and *O. niloticus amh* [90]. With the development of the technology of deep sequencing in recent years, the number of vertebrates with characterized SD genes has significantly increased, revealing that *amh* and *dmrt1* have similar functions in several species (Table 1). Yet, genes known to be involved in sex differentiation frequently adopt an MKR role in SD. These genes usually affect sex-hormone synthesis, directly or indirectly. In fish, estrogens and androgens are critical for ovarian and testicular differentiation and maintenance, respectively [17,91]. Estrogens are produced by the conversion of androgens through cytochrome P450 aromatase [92], which is encoded in teleost fish by *cyp19a1a*/*b*. The *cyp19a1 a* and *b* genes are mainly expressed in the ovary and the brain, respectively [93]. Thus, *cyp19a1a* is the key gene which is essential for estrogen synthesis and androgen depletion in the ovary. The female gene pathway determines ovarian development by up-regulating *cyp19a1a* expression [17,91]. Whereas, conversely, male-pathway genes determine testicular development by repressing *cyp19a1a* expression.

More than 10 MKRs of SD have currently been identified (Table 1). Most of these discoveries were made in fish, specifically in *Oreochromis* species (*amh*, *banf2w* and *figla-like*). With an emphasis on these species, we here propose a schematic pathway for sex-initiation and maintenance based on the major players/genes and their related functional effects. Integrating the MKR listed below, the proposed scheme is shown in Figure 1. It is noteworthy that, as shown (Table 1, Figure 1), the SD MKR role is often engaged by polymorphic downstream genes in the SD cascade (e.g., *hsd17b1*). Yet, regulatory feedback allows these genes to control upstream genes.

*Sry* and *sox3*: *Sry* is the SD gene in most mammals, and is thought to have evolved from the X linked gene *Sox3* [86,87,88,89]. This assumption was strengthened by evidence from transgenic mice lines and individual mutations in humans, which showed that *Sox3* can replace the role of *Sry* and cause sex-reversal [88,89]. In some *Oryzias* species, *sox3* plays a similar MKR role [7,38], and it has been suggested that *sox3* triggers sex by up-regulation of *gsdf* in *Oryzias* [38]. In Nile tilapia, *sox3* was recently shown to be involved in oogenesis [94]. However, there is no literature connecting it to SD in tilapia species. In mammals, *Sry* controls *Sox9* which is involved in testis development [95]. In teleost fish, *sox9* is duplicated [96] and in *O. niloticus*, *sox9b* and *sox9a* show significantly higher expression in XY gonads at 30 days post hatch (dph), and in XX gonads at 5 and 10 dph, respectively [97]. However, in medaka, *sox9b* is thought to be involved in maintenance of testis differentiation, without a role in initiating SD [98].

*Hsd17b1*: The gene *hsd17b1* was found as an SD MKR in the *Seriola* genus and in *Trachinotus anak* [50,51]. In humans, *Hsd17b1* is responsible for interconversion between the estrogen precursor estrone (E1) and the extremely potent estradiol (E2), together with other functions in steroid biosynthesis [50,99]. As in humans, in *Seriola* species, functional experiments suggested that the Z linked *hsd17b1* gene decreases the conversion of E1 to E2 [50]. The comparison of XY and XX gonads from *O. niloticus* shows that this gene is almost exclusively expressed in XX gonads from 5 to 35 days after hatching [100].

*Gsdf*: This gene is a member of the TGF-β superfamily, which is found mainly in teleost fish [101] and is an SD MKR in several fish species [26,47,48,49]. In *O. niloticus*, both over-expression of *gsdf* in XX fish [102] and its knockdown in XY fish cause sex-reversal [103]. As in medaka and rainbow trout, in *O. niloticus*, *gsdf* is distinctly and predominantly expressed in XY undifferentiated gonads; additionally, it was found to be expressed earlier than any other testis-differentiation-related gene, apart from *dmrt1* [102]. The function of *gsdf* is important for maintaining *dmrt1* expression, and inhibiting estrogen production in the gonad; whereas, *dmrt1* is thought to be upstream to *gsdf,* and controls *gsdf* expression [103]. Like *amh*, *gsdf* may affect the SMAD signaling pathway by stimulating a TGF-β receptor. However, *gsdf* may also function in a cytoplasmic protein network [104,105]. A recent study in mature stages (0.5–2.0 years) of XX *O. niloticus* concluded that similar receptors are targets of *gsdf* and *amh,* and that both genes control similar downstream genes [105]. Gene knockout of *gsdf* did not affect serum levels of E2, and probably does not control sex hormone synthesis directly. However, it has been suggested that *gsdf* may influence pituitary *fsh* gene expression in fish [105]. The expressions of *amh*, *amhr2*, *fshr*, *sox9a*, and *hsd17b1* were up-regulated in the ovaries of XX *gsdf* knockout fish [105]. However, these differences in XX gonads do not seem to reflect the differences of gene expression of XX and XY gonads which are relevant for SD. This is evident through comparison of the gene expression profiles of XX and XY gonads (at a control temperature of 28 °C), showing that *amh*, *gsdf*, *dmrt1,* and *sox9a* have higher expression levels in XY gonads than in XX gonads across a wide range of ages (20–180 dpf) [106].

*Dmrt1*: This gene is an MKR of SD in different vertebrate species, including teleost fish [53,54,55,56,57,58,59]. In Nile tilapia, *dmrt1* is involved in testicular development. Different studies in zebrafish and Nile tilapia suggest that *dmrt1* is down-regulated by *cyp19a1a* expression [24,107,108,109,110] and that in turn, in Nile tilapia, *dmrt1* represses *cyp19a1a* [111] and *foxl2* [112]. The activity of *foxl2* involves many different proteins that allow estrogen production, and influence granulosa cell activity [113,114]. In tilapia, *foxl2* up-regulates aromatase expression in vivo. Moreover, mutation of *foxl2* in XX tilapia fish decreased aromatase gene expression and serum estrogen levels [17,24]. Thus, repression of *foxl2* down-regulates the production of E2 [112]. It is also thought that part of the involvement of *dmrt1* in testis development is due to its ability to up-regulate two genes, *sox9b* and *sox30*, which are known for their involvement in testis development [96,115].

*Amh* and AMH singling: The gene *amh* and its receptor *amhr2* are known MKRs in teleost fish [63,64,65,66,67,68]. Specifically, in Nile tilapia, *amh* was found to be an MKR of SD [6,64]. In humans, *amh* is involved in hormone steroidogenesis in granulosa cells where it inhibits *cyp19a1* up-regulation by follicle-stimulating hormone (FSH) and luteinizing hormone (LH) [116]. In the medaka (hotei) mutant fish, *amh* signaling is responsible for down-regulation of *cyp19a1a* expression levels [17]. A recent study in Nile tilapia has shown that *amh* down-regulates *cyp19a1a* expression through SMAD protein phosphorylation [110]. A similar pathway was suggested in Atlantic herring (*Clupea harengus*), where a male specific gene, *bmpr1bby,* is a candidate MKR which enhances *amh* signaling and SMAD phosphorylation [52]. In Nile tilapia, *amh* is important for follicular development, and its knockdown causes a decrease in levels of LH, FSH, and E2 [117]. This seems in contrast to its function in SD, where its high expression is correlated with low *cyp19a1a* expression [6]. Nonetheless, it may function in the opposite way at the onset of SD, when the gonads’ fate is not yet determined.

*Fshr:* This gene was suggested as a candidate MKR in *M. cephalus* [9]. *Fshr* involvement in SD is well-established in fish and amphibians [9]. *Fshr* is presumed to be involved in functions that are related to SD through control of *cyp19a1* [9]. Specifically, in Nile tilapia, at very young ages (6–25 dph) *fshr* is highly expressed in XX gonads where *fshr* signaling may induce *cyp19a1a* expression [118].

*Figla and figla-like:* The gene *figla* is known to be associated with femaleness. *Figla* plays an essential role in the development and maintenance of the ovary, and in the suppression of spermatogenesis [119,120]. In *O. niloticus*, it was suggested that *figla* terminates the male gonadal differentiation, either by manipulating steroid production or by the meiotic regulation of spermatocytes [119,120]. A homolog gene, *figla-like*, is a candidate MKR in multiple tilapia species [36]. In tongue sole (*Cynoglossus semilaevis*), a *figla* homolog has a role in spermatogenesis, and in the regulation of the synthesis of steroid hormones, which are required for male determination [121]. Thus, *figla* homologs, including *figla-like*, have the potential of participating in the control of steroid hormone synthesis. *Figla-like* was down-regulated in *foxl2* knockdown fish, which suggests a role for *foxl2* in its regulation [24].

*Banf2*: A *banf2* homolog, designated as *banf2w,* was proposed as an MKR of SD in multiple cichlids [71]. It was suggested that *banf2w* regulates *foxl2* by repression of *banf1*, which in turn is a repressor of *foxl2*’s activity in gonad development [122,123,124,125].

***sdY***: In salmonids, an immune-system-related SD MKR (*sdY*) has been found [20]. The gene *sdY* encodes a protein with similarities to interferon regulatory factor 9, which mediates the interferon antiviral response [20]. However, a recent study has found that *sdY* integrates into the canonical SD cascade by interaction with *foxl2*, which is associated with ovarian development [23].

***Gdf6*:** This growth differentiation factor is an SD MKR in killifish (*Nothobranchius furzeri*) and presumably in Mexican tetra (*Astyanax mexicanus*) [60,61]. Its involvement in SD is poorly characterized; nonetheless, as with *amh* and *gsdf,* it is part of the TGF-β superfamily [60,61].

*ZkY*: The gene zinc knuckle on the Y chromosome (*zkY*) was indicated as an MKR of SD in Atlantic cod (*Gadus morhua*) [62]. However, the target of this putative RNA-binding protein and its function in SD is still unknown.

## 4. From Identification to Validation of MKRs of SD

The process of validation has been termed as “winning by points rather than knockout” [126]. This concept applies to the validation of MKRs because there is no single test or knock-out experiment that proves the functional role of a specific gene. As noted by [127] ‘the only option… is to collect multiple pieces of evidence, no single one of which is convincing, but which together consistently point to a candidate gene’. Genetic mapping has proven a powerful tool for identifying SD loci and candidate genes in many studies [72,73,75,128]. In some cases, the sex chromosome is not morphologically different from its homologue, and only a very small region is associated with sex. In such cases, analysis of multiple strains or populations with a different recombination history may fine- map the SD region towards localization of the MKR. Thus, the conservation of SD regions among multiple species with similar SD systems presents strong evidence supporting the findings of SD MKRs [7,36,71,129,130].

The actual validation of an MKR of SD is usually performed by functional studies [67,131]. The candidate gene is manipulated by different techniques and the resulting phenotype is considered as proof of function. These methods use transgenic fish manipulated by genomic editing with CRISPR/Cas9, or other methods such as TALEN and antisense RNA [6,67,131,132]. However, manipulations of genes that are part of the sex cascade often alter SD, even though they are not the causative variation. Many functional studies in Nile tilapia (*Oreochromis niloticus*), including manipulation of different genes such as *cyp19a1a* and *gsdf,* resulted in sex-reversal [24,101]. Yet, these genes are not the SD MKRs in this species [36]. Moreover, many central genes of the SD cascade, such as *sox9* and *dax1,* have so far not been identified as MKRs of SD (Table 1), although variations and manipulations of these genes indicate that they may control sex [25,133]. For example, it is unclear whether *amhy* or *amhΔy* is the MKR in Nile tilapia. Initially, genetic studies localized the SD genomic region by positional mapping [64,90,134,135], and recognized a unique variant copy of *amh* in males with a 233 bp deletion in exon 7, thus highlighting it as the candidate MKR [64]. Performed on the Swansea strain of O. *niloticus*, a follow up study showed that this form, with additional alterations, encodes a truncated protein due to an insertion of 5 bp into exon 6. This form was designated *amhΔy* [6]. In addition, this study found that *amhΔy* is in tandem with a regular *amh* gene (*amhy*) on the Y chromosome, and based on functional studies and transgenesis, it has been suggested that an SNP in exon 2 of *amhy* controls sex. According to this study *amhΔy* is a pseudogene with no clear function [6]. Later studies found that this exotic SNP is specific to the Swansea strain of Nile tilapia, and is not found in any other strains of this species that have the same SD system. However, including an insertion and deletion on exons 6 and 7, the existence and structure of *amhΔy* are conserved among different strains of *O. niloticus* [37]. Thus, it was concluded that *amhΔy* may be the initiator of SD in Nile tilapia [37].

A recent, related study failed to prove that *amhΔy* is not functional because of the high similarity of *amh*/*amhy*/*amhΔy* sequences, and the difficulty of knockdown of a whole specific copy [110]. Moreover, it was discovered that the Swansea strain possessed two MKRs, on LG1 and 23 [35,64], thus proving it is a hybrid of *O. niloticus* and *O. aureus* [36]. Hence, knockout of *amhΔy* on LG23 in some of the males from this strain may not cause sex reversal due to the existence of an additional MKR (*figla-like*) on LG1. Sequence analysis for conserved elements in hybrids can also result in false conclusions due to the intra-specific sequence variability of *amh*/*amhy*/*amhΔy* between species. The truncated *amhΔy* is expressed at sex initiation [6] and may affect expression of both *amhy* and *amh*. Generally, tandem duplicates give rise to altered expression often greater than twofold; this may be caused by different mechanisms that may involve chromatin remodeling, DNA looping, frequent transcription factor binding, or other synergistic effects, which are caused by the actual duplicated sequence, and are not fully understood [136,137]. Moreover, a large deletion in the *amhy* promoter [6,110] raises the possibility that it is not functional, and that both *amhΔy* and *amhy* act as one unit under the control of the *amhΔy* promoter. The pronounced effect of the distance between promoter elements and genes was discussed for the mammalian *sox9* [138]. In addition, there is a possibility that alternative splicing may overcome the stop code on exon 6 of the *amhΔy* gene. Thus, it has not been rejected that *amhΔy* itself may produce a functional protein. This study eventually attributed the non-functionality of *amhΔy* through an analysis of conservation in different strains [110], since a strain of Nile tilapia from Lake Koka in Ethiopia lacks the *amhΔy* 233-bp deletion on exon 7 [139]. In addition, the studies that analyzed the SD region of tilapia from the lakes in Ethiopia suggest a rapid turnover of SD loci in *O. niloticus* wild populations [139,140]. This is based on the observation that in some cases, these wild populations did not segregate for the *O. niloticus* SD locus on LG23, and in other cases, the *amh* polymorphism between males and females was different from that found in other *O. niloticus* populations [139,140]. However, detailed analysis of the *cox1* sequences from these libraries (SRA accession numbers: SRX8948078, SRX8948077 and SRX14028757), which is used as a DNA barcode standard for species assignment, revealed 2–4% differences from the *O. niloticus* barcode references. Thus, none of these libraries are bona fide *O. niloticus*, as 1% is considered the threshold that indicates species divergence in the *cox1* reference fragment [141,142,143,144,145,146]. Hence, analysis of SD in these species and populations provides important data for exotic tilapias, but is not relevant for the study of SD initiation by *amhΔy* in *O. niloticus*. Moreover, in recent studies, we showed cases of taxonomic classification errors of tilapia species [36,147]. Therefore, although multiple functional studies have been performed, it is not yet clear what the causative polymorphism of SD on LG23 of Nile tilapia is, and most evidence is still based on the conservation of elements in the SD locus among multiple strains. Considering all the evidence, it seems that the structural variation of *amhΔy* may have an MKR role by inducing over-expression of the *amhy* gene that affects the estrogen/androgen equilibrium.

## 5. The Significance of Taxonomic Authentication of Species Using DNA Barcoding for MKRs Search

The identification of taxonomy errors is common in the tilapia species as there are many species in this genus that are frequently hybridized for aquaculture purposes and then escape to the wild, affecting native tilapia populations in natural water resources [148,149]. Mass production of tilapia is based on hybridization of two or more tilapia species, usually *O. niloticus*, *O. aureus*, *O. mossambicus*, and *O. urolepis hornorum,* in which three different MKRs govern SD. The MKRs of *O. niloticus* and *O. mossambicus* were mapped on LGs 23 and 14, respectively [37,150]; both *O. aureus* and *O. urolepis hornorum* have the same MKR i.e., *banf2w* on LG3 [30,71,151,152]. Consequently, commercial strains may segregate for the original and the de novo SD loci which emerged following hybridization [36,37,147]. This additional complexity of SD of hybrid species stems from the confluence of different alleles of loci which had no effect on SD in their species of origin. As a result of hybridization, these alleles may segregate and influence SD. The first generations of such hybrids, which are reared in commercial ponds, have strong advantages due to fast growth, high reproductive levels, and excellent adaptation to a wide range of temperatures and salinities in rearing ponds [153,154,155]. However, having high invasive characteristics, hybrids are less affected by natural reproductive barriers between species, and therefore endanger native tilapia populations [154,156]. Different explanations have been given for the decline in size of tilapia populations in natural sources which have been polluted by commercial hybrids; these include the transfer of pathogens and parasites from ponds, weaknesses of commercial traits in the wild, and accumulated chromosomal aberrations driven by faulty crossing-over of homologous chromosomes [148].

In vertebrate mitochondrial oxidative phosphorylation, 13 mitochondrial proteins have evolved with more than 70 nuclear-coding proteins of inner mitochondrial membranes [157]. These proteins cooperate in ATP production through oxidative phosphorylation co-adapting during evolution [158]. Consequently, there is an expected variability of mitochondrial and nuclear sequences, even between closely related species such as *O. niloticus* and *O. aureus*, which can be attributed to the differentiation of the species [159,160]. Thus, reports of discordance between genomic and mitochondrial sequence variants in *O. niloticus* and *O. aureus* tilapia populations suggest that hybridization has been caused by relatively recent aquaculture pollution [148,149]. In our opinion, the destructive influence of commercial activity might have given rise to the erroneous scientific hypothesis of “rapid turnover” of MKRs of SD [140], which in turn may have legitimized negative commercial activity. Thus, it is essential that the taxonomy of a species is initially verified before studying its SD mechanism. The BOLD taxonomy system provides an infrastructure for such preliminary tests. The latest progress in DNA recovery from formalin-fixed samples [161] could be utilized to examine these hypotheses through the analysis of the thousands of cichlids, collected and preserved over the last 120 years, which are kept in numerous museums, private collections, and laboratories.

## 6. Single vs. Polygenic SD Systems

Currently, most of the findings in species with a genetic SD system show that the sex trait is categorical, and controlled by a single gene (Table 1). Moreover, a single-factor SD system is not only limited to species where a sex-biased ratio is considered disadvantageous for its fitness. This phenomenon could be explained by genomic conflicts between parents and offspring [162], and is demonstrated by closely related species of the *Oryzias* genus, which have seven different SD genes, even though SD in each species is controlled by a single MKR [7].

Nonetheless, there are cases with polygenic SD systems. This is especially common in commercial and laboratory stocks, which may differ from the natural population which segregates for a stable mono-factorial SD system. In zebrafish (*Danio rerio*), domesticated strains have lost the natural SD region on chromosome 4 [163]. In fighting fish (*Betta splendens*), it was suggested that a polygenic SD system evolved through hybridization and selection, and that the complexity of SD in different populations is dependent on the chances of admixture [164]. Laboratory stocks of guppy, defined as *Poecilia reticulata,* are possibly hybrids of two species, *P. reticulata* and *P. wingei* [81]. In contrast to tilapia species, both *Poecilia* species have morphologically different sex chromosomes with a common region on LG12; consequently, their hybrids still segregate for a single SD locus with varying recombination patterns in the two strains [82,165].

Although the loss of stability of the SD systems in commercial stocks is common, the evidence of novel sex chromosome systems following hybridization is rare because of slow evolutionary processes [166]. A 30-year experiment on swordtail fish demonstrated how the hybridization of two different species can cause the evolvement of a new SD system on a new chromosome [166]. In this example, the SD system was translocated from one of the species to a new genetic locus in the examined hybrid.

In tilapia, multiple SD MKRs have frequently been found in different families of the same species [36]. However, tilapia aquaculture involves hybridization, and therefore commercial stocks of fish may present SD systems governed by several segregating loci as a result of these practices [27,36]. Nile tilapia has an SD system on LG23 which is controlled by the *amh* gene [6,37,64]. However, different studies have found that Nile tilapia also segregates for an SD system on LG1 [35]. A recent study showed the hybrid origin of two stocks with an LG1 SD system, and explained how LG1 evolves as an SD gene in these hybrids [36]. It was suggested that SD MKR on LG1 is lacking in Nile tilapia, whereas it is autosomal in *O. aureus*. Thus, a cross of the two species causes segregation of this locus in the same way as a natural SD MKR of an XY SD system [36]. This can explain how the mono-factorial SD system on LG23 of purebred *O. niloticus* dispersed to additional loci on LGs 1 and 3 in hybrid strains, such as Chitralada, Amherst, Stirling, and Swansea [35,36,37,147]. Thus, mitochondrial-nuclear genome concordance, and mono-factorial SD systems are important indicators of the purebred origin of stocks and populations. Multiple SD loci, which have been reported for other African cichlid species [83,167], need to be further validated by testing if these are of purebred origin.

## Figures and Tables

**Figure 1 ijms-24-02468-f001:**
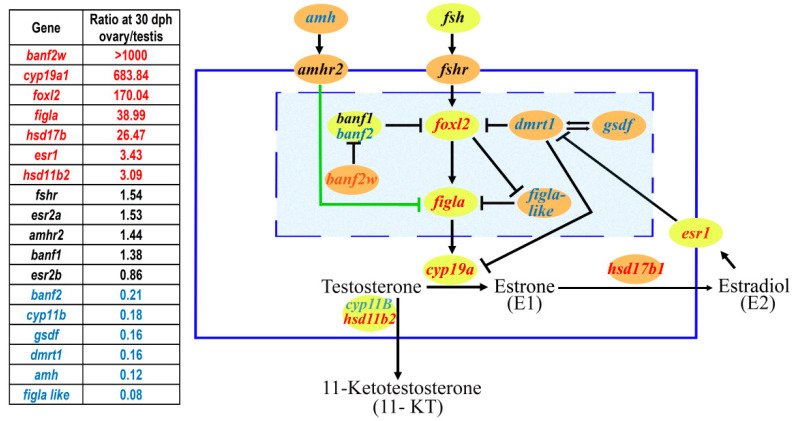
A proposed schematic model of the sex determination (SD) pathway in tilapia, based on current knowledge of master key regulators (MKRs) of SD and their effects on estrogen/androgen (E2/11-KT) equilibrium. Identifications of MKRs of SD from different fish species are colored orange and other factors are yellow. The table on the left indicates the RPKM (Reads Per Kilobase of transcript per Million mapped reads) of ovary/testis expression-values of genes in *Oreochromis aureus* at 30 days post hatching (dph) (BioProject accession number: PRJNA609616). Genes with male and female biased expression values are colored in blue and red, respectively. Arrows (↓) represent up- regulation/production, whereas blocked arrows (⊥) mark down-regulation/production. The green lines represent the SMAD signaling pathway. MKRs of SD without a clear function or relevance to SD in tilapia are not displayed (*gdf6*, *sdY, dmy, sox3*, *bmpr1bby* and *zkY*) in the figure. However, *gdf6* and *bmpr1bby* are members of the TGF-beta superfamily and may affect SMAD signaling, and thus may have similar functions to those of *amh* and *gsdf*. Other undepicted SD MKRs are the *sdY* gene that interacts with *foxl2* in Salmonids, *sox3* that regulates *gsdf,* and *zkY* with unknown functions. The blue solid line represents the cell membrane.

## Data Availability

Not applicable.

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
