# Peer review of "Master-Key Regulators of Sex Determination in Fish and Other Vertebrates—A Review"

_ijms, 2023, doi:10.3390/ijms24032468_

Round 1
Reviewer 1 Report
The authors declared they summarized the master-key regulators of sex determination in fish and other vertebrates. It is important for understanding the sex determination mechanism and searching for sex determination related genes. According to my known literatures, some of the genes related to sex determination were not listed, such as dmY, Sxl, fem, csd, fgf9, fgfr2, and β-catenin etc. The discussion of these genes' relationship is inadequate (Fig. 1).
1. The full name should be given when the gene name appears for the first time
2. The font size of gene name in Figure 1 is too small to read
3. Tilapia is too much discussed to represent most fishes in nature
4. Gonosomes is not recommended. Instead, using sex chromosomes
Author Response
Review Report Form 1
Comments and Suggestions for Authors
The authors declared they summarized the master-key regulators of sex determination in fish and other vertebrates. It is important for understanding the sex determination mechanism and searching for sex determination related genes. According to my known literatures, some of the genes related to sex determination were not listed, such as dmY, Sxl, fem, csd, fgf9, fgfr2, and β-catenin etc. The discussion of these genes' relationship is inadequate (Fig. 1).
As indicated also by the review title, we focus on master-key regulators (MKRs) of sex determination (SD) of vertebrates, whereas the SD MKRs indicated by the reviewer originate mostly from insect research: Sxl in fly (Drosophila melanogaster); fem in worm (Caenorhabditis elegans); csd in honey bee (Apis mellifera L.). We did listed dmrt1Y, which is synonymous with dmY (see https://www.ncbi.nlm.nih.gov/gene/101161472). The other genes are important for the sex cascade; yet, they are not considered as initiators and thus are not suggested as SD MKRs. Likewise, there are hundreds of genes that affect the sex cascade but are not considered as MKR in this review. Nevertheless, as this distinction is important, we incorporated this explanation in lines 41-43 "However, this review considers only those candidate MKRs that were suggested after isolating a specific DNA sequence variation, which was then confirmed by recombination boundaries, multiple species conservation, or functional studies".
- The full name should be given when the gene name appears for the first time
In current literature, the gene symbol is considered as the proper gene name without providing further explanation like for acronyms. Indeed, full gene names can consist of as much as ten words and therefore using them in text sentences might make them unreadable. To avoid this, we provide the full names as information in Table 1, which is presented early in the manuscript.
- The font size of gene name in Figure 1 is too small to read
We increased the font size of all gene names by 130-140%.
- Tilapia is too much discussed to represent most fishes in nature
Because of the abundance and importance of tilapia to aquaculture, this clade has been used in relatively high number of genetic studies in comparison to other fish. These studies have explored genetic markers, built dense linkage maps, pursued genome-wide association studies for genes of economic importance, produced gene expression data and functional experiments using genome editing. The incentive to produce all-male populations of tilapia has driven the research to elucidate its SD mechanism and investigate the diversity of SD genes and their functional characteristics. This indeed may have caused an imbalanced data considering the wide variety of fishes in nature. However, we believe that sharing our insights of tilapia SD is reasonable; and we hope that future research would provide data for many additional fish species to be included in a wider review.
- Gonosomes is not recommended. Instead, using sex chromosomes
We replaced "gonosomes" with "sex chromosomes".
Reviewer 2 Report
This manuscript “Master-key regulators of sex determination in fish and other vertebrates - a review” by Arie Yehuda Curzon, Andrey Shirak, Micha Ron and Eyal Seroussi summarised a list of 14 Master key regulators (MKRs) in fish and vertebrates, and reviewed the searching, identification, and verification of these genes as MKRs. The authors illustrated a network that affects estrogen/androgen equilibrium by these MKRs integrated with other sex determining genes. Significantly, the authors examined sex specific mortality, sex reversal, environmental impacts on sex determination, multigenic sex determination and point out these phenomena in fish are often consequences of artificial hybridization, adding complexity in search and analysing MKRs. The authors also discussed the essentiality necessary to validate purebred origin before MKRs searches. In addition, the authors suggested the heterologous sequences of these MKRS could be used as targets in future searches for MKRs in additional species owing to limited number of conserved MKRs.
The review was clearly written, comprehensive and relevant to the field of Master Key Regulators and sex determination. The figure and table are appropriately prepared and easy to understand. Statements and conclusions are reasonably drawn and supported by the citations.
Particularly, this review assesses the complex impacts of fish artificial hybridization on segregation and recombination, translocations of sex determination loci, and evaluate the significance of taxonomic authentication of species for MKRs search. This is very applicable for searching MKRs in fish species.
As I know, no similar review was published recently. The references are most recent, relevant, and appropriately cited. No relevant citations omitted. no excessive number of self-citations.
The review is acceptable, but with minor improvements required.
Questions or comments concerned as followed:
1. In section 3. Identification of SD MKRs,
some MKRs such as Hsd17b1 in seriola genus, Gsdf in teleost fish were not mentioned on which sex chromosomes or SD loci they are linked. I recommend to add the information for clearity.
2. Line 79
“that were not known to be involved in SD,” make the meaning confused. It is better to delete it.
3. In line 197 to 202
‘dmrt1 is thought to be upstream to gsdf,’ while gsdf not dmrt1 is regarded as MKR in teleost fish. Why? Is that because dmart1 is not linked to sex chromosome or SD region but gsdf is?
4. Line 62. “ZW/WW’ should be ZW/ZZ?
5. Line 317 and 318
“an SNP” should be “a SNP”
Author Response
Review Report Form 2
Comments and Suggestions for Authors
This manuscript “Master-key regulators of sex determination in fish and other vertebrates - a review” by Arie Yehuda Curzon, Andrey Shirak, Micha Ron and Eyal Seroussi summarised a list of 14 Master key regulators (MKRs) in fish and vertebrates, and reviewed the searching, identification, and verification of these genes as MKRs. The authors illustrated a network that affects estrogen/androgen equilibrium by these MKRs integrated with other sex determining genes. Significantly, the authors examined sex specific mortality, sex reversal, environmental impacts on sex determination, multigenic sex determination and point out these phenomena in fish are often consequences of artificial hybridization, adding complexity in search and analysing MKRs. The authors also discussed the essentiality necessary to validate purebred origin before MKRs searches. In addition, the authors suggested the heterologous sequences of these MKRS could be used as targets in future searches for MKRs in additional species owing to limited number of conserved MKRs.
The review was clearly written, comprehensive and relevant to the field of Master Key Regulators and sex determination. The figure and table are appropriately prepared and easy to understand. Statements and conclusions are reasonably drawn and supported by the citations.
Particularly, this review assesses the complex impacts of fish artificial hybridization on segregation and recombination, translocations of sex determination loci, and evaluate the significance of taxonomic authentication of species for MKRs search. This is very applicable for searching MKRs in fish species.
As I know, no similar review was published recently. The references are most recent, relevant, and appropriately cited. No relevant citations omitted. no excessive number of self-citations.
The review is acceptable, but with minor improvements required.
Questions or comments concerned as followed:
- In section 3. Identification of SD MKRs,
some MKRs such as Hsd17b1 in seriola genus, Gsdf in teleost fish were not mentioned on which sex chromosomes or SD loci they are linked. I recommend to add the information for clearity.
We added into Table 1, the chromosomal locations of the SD genes indicated by the relevant references.
- Line 79
“that were not known to be involved in SD,” make the meaning confused. It is better to delete it.
Done.
- In line 197 to 202
‘dmrt1 is thought to be upstream to gsdf,’ while gsdf not dmrt1 is regarded as MKR in teleost fish. Why? Is that because dmart1 is not linked to sex chromosome or SD region but gsdf is?
A gene that is not polymorphic between the sexes cannot initiate sex determination and therefore is not considered as master key-regulator in this review. This role is often taken by polymorphic downstream genes. Yet, regulatory feedback allows these genes to regulate upstream genes. We incorporated this explanation in lines 174-176 "It is noteworthy that as shown (Table 1, Figure 1) the SD MKR role is often taken by polymorphic downstream genes in SD cascade (e.g., hsd17b1). Yet, regulatory feedback allows these genes to regulate upstream genes."
- Line 62. “ZW/WW’ should be ZW/ZZ?
Done.
- Line 317 and 318
“an SNP” should be “a SNP”
Acronyms beginning with a consonant sound use “a”. Acronyms beginning with a vowel sound use “an”. Thus pronouncing "an es-en-pee" is valid, yet, pronouncing "a snip" is also common. We prefer the former. This is well explained here: https://www.scribbr.com/commonly-confused-words/a-vs-an/